# DEPTH FROM CAMERA MODEL

## ABSTRACT

Depth estimation is a critical topic for robotics and vision-related tasks. In monocular depth estimation, However, the current accuracy of supervised learning needs to be improved, and the generalization of supervised learning models to other scenarios is poor, making it difficult to directly estimate depth for other scenarios. In deep learning era, while existing methods mainly rely on the exploration of image relationships to train the supervised neural networks, fundamental information provided by the camera itself has been generally ignored, which can provide extensive supervision information for free, without the need for any extra equipment to provide supervision signals. Utilizing the camera itself's intrinsics and extrinsics, depth information can be calculated for ground regions and regions connecting ground based on physics principles, providing free and critical prior information without any other sensors. The method is easy to realize and can be a component to enhance the effects of all the supervised methods.

## 1    INTRODUCTION

Monocular Depth Estimation (MDE) serves as a cornerstone in robotics and computer vision, having a broad spectrum of applications from autonomous driving to augmented reality Tang et al. (2022) and 3D reconstruction Newcombe et al. (2011). At its core, MDE strives to assign a depth value to every pixel in an image. The advent of convolutional neural networks has propelled the field of MDE to new heights Simonyan & Zisserman (2014); Szegedy et al. (2015); He et al. (2016). Presently, various convolutional neural network architectures Eigen et al. (2014); Fu et al. (2018); Lee et al. (2019) have exhibited prowess in MDE. Furthermore, with the rise of VIT Dosovitskiy et al. (2020), transformer-based approaches Ranftl et al. (2021); Yang et al. (2021); Yuan et al. (2022) are gaining traction. Notably, these methodologies predominantly focus on training models to predict depths from single images utilizing ground truth. However, the inherent ambiguity of MDE poses challenges, as MDE is theoretically an ill-posed problem, which can mainly estimate depth based on learned scenes. Due to the missing second camera or active sensors, MDE systems trained on one type of scene (e.g., outdoor) typically do not perform well on other types of scenes (e.g., indoor). Current deep learning paradigms, despite their advancements, have not addressed this underlying issue, leading to limitations in their generalization capabilities.

Paradoxically, in the era dominated by deep neural networks, pivotal sensor-specific information often gets overshadowed. In this paper, we introduce a seminal approach that capitalizes on camera model parameters, both intrinsic and extrinsic, to compute scene depth, thereby providing a solid depth prior for depth estimation training and inference. Leveraging these camera parameters, we can ascertain depth across substantial portions of a scene with remarkable accuracy, enabling effective neural network training without explicit ground truth dependencies. Through semantic segmentation, ground plane depth can be deduced based on the camera's physics model. This foundational depth estimation facilitates the subsequent computation of depths for objects (e.g., buildings, vehicles) situated on this plane. This mechanism effectively provides depth input priors at no additional cost beyond the camera's inherent capabilities. Incorporating this physics-based depth, we devise unique fusion modules to amalgamate physics depth with RGB imagery, serving as inputs that synergize with networks. More crucially, this strategy can seamlessly integrate with any supervised depth estimation framework.

In summary, our contributions are threefold: 1. We introduce a groundbreaking mechanism that harnesses the camera's physics model parameters to compute scene depth, which we term as *physics depth*. 2. Our proposed information fusion module adeptly integrates physics depth into image

data, yielding multi-modal features. This enriched output can subsequently feed into any supervised model, markedly enhancing depth prediction accuracy. 3. A simple yet effective network training system providing attention to two different input modalities (RGB and physics depth) for feature extraction and fusion has been proposed, which is validated for depth from camera model framework. 4. We present a methodology to both validate and rectify the calibration results of camera orientation relative to the ground.

## 2 RELATE WORK

### 2.1 MONOCULAR DEPTH ESTIMATION

**Supervised model:** In the domain of Monocular Depth Estimation (MDE) using neural networks, the seminal work of Eigen et al. (2014) constitutes a pivotal and foundational contribution. Their research introduces a coarse-to-fine convolutional neural network. On the basis of his model, a range of methods have emerged. It can be categorized into two distinct directions: one that involves the monodepth estimation problem as a pixel-wise regression task Huynh et al. (2020), and another that formulates it as a pixel-wise classification challenge Cao et al. (2017). In the decoding stage, Lee et al. (2019) introduced a local planar guidance layer to infer the plane coefficients, which were used to recover the full depth of resolution of the map. More recently, Adabins Bhat et al. (2021) has applied transformer to estimate the depth.

**Self-supervised model:** Self-supervised depth estimation from monocular videos or stereo image pairs is emerging due to the mitigation of manual labeling efforts. In the realm of monocular depth, Zhou et al. Zhou et al. (2017) pioneered a self-supervised framework. This was achieved by jointly training depth and pose networks anchored on an image reconstruction loss. Subsequently, Godard et al.Godard et al. (2019) established a benchmark through the proposal of a minimum re-projection loss and auto-masking loss. Building upon the foundation laid by Godard et al. Newcombe et al. (2011), several studies Guizilini et al. (2020); Chawla et al. (2021) addressed the inherent scale ambiguity of monocular SfM-based methods through integration with other sensors.

**MDE network architecture:** Monocular depth estimation performance varies significantly across different architectures. Yin & Shi (2018) transitioned from the VGG encoder to a ResNet encoder. Guizilini et al. (2020) introduced 3D convolutions in PackNet, aiming to efficiently compress and decompress features while preserving details. To fuse multi-scale features, Wang et al. (2020) employed attention mechanisms. Recognizing the inherent limitations of CNNs, Zhou et al. (2021) integrated HRNet for self-supervised monocular depth estimation, capitalizing on HRNet's Wang et al. (2020) prowess in modeling multiscale features. In the latest time, Transformer is also developed for depth estimation Li et al. (2023b).

### 2.2 GEOMETRIC PRIORS

Geometric priors have gained traction in the domain of monocular depth estimation. Among these, the normal constraint Long et al. (2021); Qi et al. (2018)—which enforces consistency between the normal vectors inferred from estimated depths and their ground truth counterparts—is prevalent. The piecewise planarity prior Chauve et al. (2010); Gallup et al. (2010); Bódis-Szomorú et al. (2014) offers a tangible approximation to real-world scenarios. This prior segments the scene into 3D planes Yang & Zhou (2018); Zhang et al. (2020), aiming to categorize the scene into dominant depth planes.

Notwithstanding the inherent ambiguity in monocular depth estimation, contemporary supervised learning paradigms remain predominantly grounded on truth labels. Even as novel architectures like the Transformer enhance prediction accuracy, they do not address the foundational challenges associated with monocular depth estimation errors. While geometric priors can mitigate some uncertainty, their contributions to the overarching problem remain marginal. Diverging from traditional geometric priors, we leverage camera model parameters to compute scene depth. This approach furnishes more precise and generalizable depth predictions, largely bolstering model performance.

## 3 PHYSICS DEPTH COMPUTATION BASED ON CAMERA MODEL

In this work, we introduce a methodology to compute the depth of flat surfaces, particularly those directly within the camera's field of view, denoted as *physics depth*. This approach is especially beneficial for estimating road depths using ground robot cameras. By harnessing both intrinsic and extrinsic camera param-

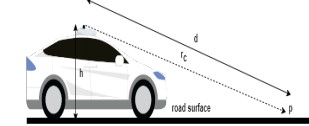

Figure 1: Illustration of physics depth logic for a car moving on a flat road surface

eters, combined with semantic segmentation using models like Zhu et al. (2019), we identify ground surfaces within images. Furthermore, we determine regions adjoined to these surfaces and compute their depth values. Evaluations on the KITTI dataset Geiger et al. (2013) corroborate the accuracy of our method, revealing close alignment with LiDAR-derived depth for proximate road surfaces. Our analysis also uncovers calibration inconsistencies in the KITTI dataset spanning multiple recording days, and we detail a correction method for the camera-to-ground calibration matrix. We have extended our depth estimation technique to cater to surfaces with minor undulations, with outcomes elucidated subsequently.

## 3.1 GROUND SURFACE PHYSICS DEPTH COMPUTATION

The *physics depth* pertains to pixels representing an ideal flat surface in the camera's direct line of sight. For illustrative purposes, we consider a standard pinhole camera, commonly devoid of significant distortion, as our model. Nevertheless, our approach is adaptable to other camera varieties with appropriate model modifications. Utilizing the image's semantic segmentation, we differentiate flat terrains from other surfaces. Given the precision of semantic segmentation in standard scene delineation, it is deployed as established features like SIFT. For each pixel indicative of a frontal flat surface, we derive a unit vector $r$, indicative of the camera ray's direction in the physical realm.

$$r = \frac{[u,v,f]}{\sqrt{u^2+v^2+f^2}}, \quad f = [f_x + f_y]/2 \tag{1}$$

where $u$ represents the coordinates of the pixel, with the origin of the coordinate system situated at the optical center of the image, commonly referred to as the principal point. Meanwhile, $f_x, f_y$ denote the camera's focal lengths in the $x, y$ directions. Then, we rotate the camera ray vector to align it with the ground plane:

$$r_c = R_c r \tag{2}$$

Here, $R_c$ is a 3x3 rotation matrix representing the camera's orientation relative to the ground coordinate system. Using the camera's roll and pitch angles, the rotation matrix can be computed as illustrated below:

$$R_c = R_{yaw} * R_{pitch} * R_{roll} \tag{3}$$

Since $r_c$ is a unit vector, the 3D coordinates of a ground point relative to the camera's coordinate system can be determined by multiplying it with the depth $d$.

$$p = d * r_c, d * r_c = [p_u, p_v, p_f], \quad d = \frac{p_u}{r_{c,u}} = \frac{p_v}{r_{c,v}} = \frac{p_f}{r_{c,f}} \tag{4}$$

where $p = [p_u, p_v, p_f]$ is the ground point relative to the camera coordinate system, $d$ is the distance from the camera to the ground point and $r_c = [r_{c,u}, r_{c,v}, r_{c,f}]$ is a unit vector in the direction of camera ray in physics world. Therefore, $p_v$ must be equal to camera height $h$. For example, in KITTI camera coordinate system, v direction points downward and camera height, $h$ is 1.65. Therefore, depth d can be calculated as below:

$$d = \frac{h}{r_{c,v}} \tag{5}$$

Figure 1 illustrates the physics depth logic for an autonomous vehicle with a camera on top of it. Here, h is the height of the camera, d is the distance of the camera from point p and rc is the direction of the camera ray towards the point p in Fig. 1. For every pixel $u, v$ that corresponds to a ground point, we repeat the above algorithm to calculate the depth. Therefore, we can calculate the depth value of every ground surface pixel, as Fig. 2. The error for the calculated physics depth is quite small compared with LiDAR ground truth.

## 3.2 EXTENSION OF GROUND PHYSICS DEPTH

In our evaluations, the physics depth aligns closely with LiDAR measurements for most scenarios, providing a dense depth map as opposed to the sparser LiDAR counterpart. However, this approach predominantly targets the road directly ahead of the robots, which could lead to overfitting to road regions when training a depth prediction model, restricting its applicability to non-road surfaces.

To mitigate this overfitting risk, we broadened the scope of our physics-based depth method to cover the entire ground plane—encompassing roads, sidewalks, parking lots, rail tracks, and more—by presuming a uniform flatness over these terrains. Furthermore, we extrapolated this depth approach to vertical entities like vehicles, pedestrians, and buildings. This extension was achieved by propagating depth values vertically from the intersection of horizontal and vertical structures. With these enhancements, we can now represent nearly 70% of all image pixels with an error margin below 20%, and over half of the image pixels has error less than 5%.

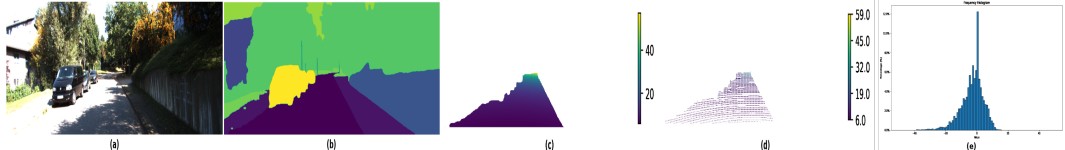

Figure 2: **Physics Depth Algorithm on a sample KITTI image:** This figure includes: (a) an RGB image of the scene, (b) the corresponding semantic segmentation image, (c) the physics depth map for road pixels, (d) the LiDAR depth map for road pixels, and (e) the percentage error frequency distribution for road pixels, comparing values between physics depth and LiDAR depth.

### 3.3 CORRECTION OF CAMERA-TO-GROUND CALIBRATION

Camera extrinsic is an important component in calculating the physics depth. Extensive methods have been developed for camera calibration, such as Zhengyou (1998). In our analysis of the Physics Depth Algorithm on the complete KITTI dataset, optimal results were evident on the initial day (2011-09-26). However, performance diminished in the subsequent days. Here, we provide a camera-to-road calibration verification and correction method. Taking KITTI dataset as an example, given the algorithm's excellence on day one, we postulated that discrepancies in the later days might stem from inconsistencies in the KITTI dataset's camera calibration parameters, specifically $R_c$.

To substantiate this, we aimed to compute the camera calibration rotation matrix $R_c = R_{cl} \times R_{lg}$ using LiDAR data. Here, $R_{cl}$ is the rotation matrix to transition a point from the camera to the LiDAR frame, and $R_{lg}$ transforms a point from the LiDAR to the ground frame. Below is the procedure for verifying and correcting the camera-to-road calibration.

**Camera Calibration Correction Methodology:**

1. $R_{lc}$ is provided in the KITTI's calibration set. Compute $R_{cl}$ as the transpose of $R_{lc}$.
2. Derive $R_{lg}$ via:
   (a) Project 3D depth points from the LiDAR onto the LiDAR frame to form a 3D hyperplane of the ground.
   (b) Determine the centroid of these 3D ground points in the LiDAR frame. Decompose the surface normal at this centroid to ascertain roll and pitch angles.
   (c) Refer to the third step of the physics depth algorithm to obtain $R_{gl}$ using the derived angles.
   (d) Compute the transpose of $R_{gl}$ to yield $R_{lg}$.
3. Combine $R_{cl}$ and $R_{lg}$ from the above steps to produce $R_c$.
4. Utilize $R_c$ in the third step of the physics depth algorithm to achieve the Camera Calibration Corrected Physics Depth.

Note that robotics calibration is usually precise. Here, we are providing a method to verify and correct the camera-to-road calibration output. Through our testing, the camera-to-road calibration in KITTI dataset maintains errors, which we will demonstrate and provide the corrected results in the experiment part.

## 4 INFORMATION FUSION MODULE

In this section, we introduce a comprehensive Information Fusion module, designed to merge our physics depth with RGB images, as illustrated in Fig. 3. The module encompasses three integral parts: MHRA (Multi-Head Relational Attention), Depth Information Selection, and the Supervision Model. This fusion strategy aims to leverage the interplay between physics depth and traditional image information, enhancing the supervised learning model's performance in monocular depth estimation.

## 4.1 FUSION UNIFIED FRAMEWORK

As depicted in Fig. 3, our Fusion Unified framework is inspired by the spatio-temporal network structure of UniFormer Li et al. (2023a). This design extracts and integrates features from RGB and depth images to serve as inputs for supervised learning models. Our approach recognizes that while physics depth provides partial depth information for an image, a relationship exists between these depths and other image regions. By learning this mapping, our network can better handle monocular depth estimation tasks, leveraging the advantages of the physics depth.

The Fusion Unified framework comprises five pivotal modules: Dynamic Position Embedding (DPE), global and local Multi-Head Relation Aggregator (MHRA), Physics Depth Selection (PDS), and the Supervision model. Given the accuracy of road depth data, the PDS module introduces an adaptive weighting mechanism to the physics depth features, determining the emphasis of each feature and mitigating overfitting risks.

In the MHRA module, our goal is to capture both global and local details from the physics depth and RGB images. While the road depth data is precise, an over-reliance on the physics depth can lead to overfitting. Consequently, it is vital to judiciously leverage this depth information. In the PDS module, we introduce a weight matrix for the physics depth features, determining which aspects to emphasize and which to downplay. Given that these weight matrices are learned, the network can adaptively modulate the inclusion of depth information, ensuring it does not solely rely on the depth features.

### 4.1.1 GLOBAL AND LOCAL MULTI-HEAD RELATIONAL ATTENTION

The MHRA module aims to extract both global and local information from the physics depth and RGB images. While CNN structures excel in capturing local information, Transformer architectures, due to the attention mechanism, adeptly extract global details. In our MHRA design, local MHRA targets the extraction of local RGB nuances, as RGB images contain extensive details, while global MHRA focuses on capturing the overarching trends within the physics depth data, depicting the scene structures.

$$X_{RGB} = DPE(x_{rgb}) + x_{rgb}, X_{Depth} = DPE(x_{depth}) + x_{depth} \tag{6}$$

$$Y_{RGB} = MHRA_{local}(Norm_{BN}(X_{RGB})) + X_{RGB} \tag{7}$$

$$Y_{Depth} = MHRA_{global}(Norm_{LN}(X_{Depth})) + X_{Depth} \tag{8}$$

MHRA exploits token relationships in a multi-head style:

$$R_n(X) = A_n V_n(X) \tag{9}$$

$$MHRA(X) = Concat(R_1(X); R_2(X); \cdots R_N(X))U \tag{10}$$

$R_n(\cdot)$ is the $n_{th}$ header, U is a learnable parameter matrix for the integration of N heads. $V_n(X)$ is a linear transformation for original tokens. $A_n$ is token affinity learning, which has local and global modes. For RGB images, our goal is the extraction of local information.

$$A_n^{local}(X_i, X_j) = a_n^{i-j} \tag{11}$$

$a_n$ is the learnable parameter, $(i-j)$ denotes the relative position between token i and j. The entire depth image is integrated to extract the global relationship. $Q_n(\cdot), K_n(\cdot)$ are two different linear transformations.

$$A_n^{global}(X_i, X_j) = \frac{e^{Q_n(X_i)^T K_n(X_j)}}{\sum_{j' \in \Omega_{H \times W}} e^{Q_n(X_i)^T K_n(X_{j'})}} \tag{12}$$

### 4.1.2 PHYSICS DEPTH SELECTION

In terms of depth information, the road data is highly accurate as it derives from meticulously calculated camera parameters. This module takes as inputs the road data, the filled depth information (representing physical depth), and the original RGB image. While the road information contributes precision to the depth values, the process of filling the environment with this data lacks accuracy. To rectify this, we employ a deep neural network to refine and enhance it. However, the depth information itself remains predominantly accurate, serving as a foundational global reference for our focus on physical depth. Concurrently, we prioritize the RGB information to complement the incomplete aspects of the physical depth information and rectify any discrepancies introduced during the filling process. Consequently, our emphasis lies on extracting RGB texture details, analyzing color space distribution, and other local data to refine the final output.

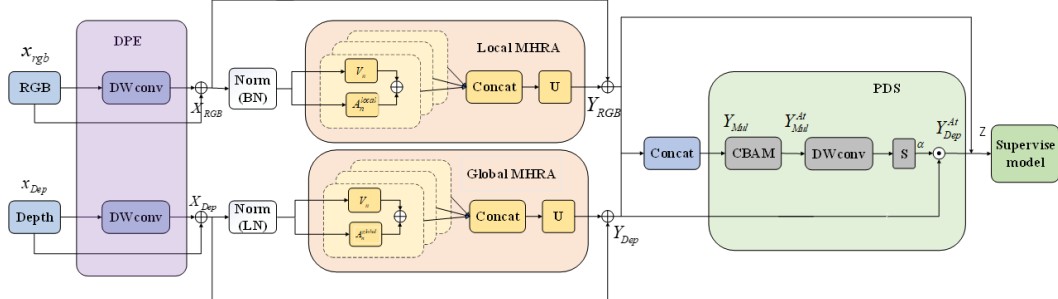

Figure 3: Information Fusion module consists of four key modules, Dynamic Position Embedding (DPE), global and local Multi-Head Relation Aggregator (MHRA), Physics Depth Selection, and the Supervision Model.

In the PDS module, we first concatenate the depth features $Y_{Dep}$ and image features $Y_{RGB}$ together, capturing multimodal features $Y_{Mul}$. $Y_{Mul}$ is then further characterized by key multimodal features $Y_{Mul}^{At}$ using the CBAM module.

$$Y_{Mul} = Y_{RGB} + Y_{Dep} \tag{13}$$

The CBAM module is Convolutional Block Attention Module Woo et al. (2018), which applies channel and spatial attention modules sequentially. This allows the module to learn "what" and "where" to attend in the channel and spatial axes, respectively. As a result, by learning what information to emphasize or suppress, our module efficiently adjusts the features within the network.

$$F' = M_c(F) \otimes F, F'' = M_s(F') \otimes F' \tag{14}$$

$$
\begin{aligned}
M_c\left(F\right) &= \sigma\left(MLP\left(AvgPool\left(F\right)\right) + MLP\left(MaxPool\left(F\right)\right)\right) \\
&= \sigma\left(W_1\left(W_0\left(F_{avg}^c\right)\right) + W_1\left(W_0\left(F_{\max}^c\right)\right)\right)
\end{aligned} \tag{15}
$$

$$
\begin{aligned}
M_s(F) &= \sigma\left(f^{7\times7}\left(\left[AvgPool\left(F\right); MaxPool\left(F\right)\right]\right)\right) \\
&= \sigma\left(f^{7\times7}\left(\left[F_{avg}^s; F_{\max}^s\right]\right)\right)
\end{aligned} \tag{16}
$$

$\otimes$ denotes element-wise multiplication; $\sigma$ denotes the sigmoid function. MLP stands for a multi-layer perceptron, where $W_0$ and $W_1$ represent shared weights for both inputs. The ReLU activation function is applied after $W_0$. $f^{7\times7}$ represents a convolution operation with a filter size of $7 \times 7$. $F_{avg}^s$ and $F_{\max}^s$ denote average-pooled features and max-pooled features, respectively.

$Y_{Mul}^{At}$ has a significant impact on the output and influences the extent to which the physical depth information is integrated into the network. To achieve this, we utilize depthwise separable convolution along with the sigmoid function to enable the network to learn the weight matrix associated with the physical depth feature $\alpha$. The formula is presented below:

$$Y_{Mul}^{At} = CBAM(Y_{Mul}), \alpha = \sigma(DW(Y_{Mul}^{At})) \tag{17}$$

$$CBAM(F) = M_s(M_c(F) \otimes F) \otimes (M_c(F) \otimes F) \tag{18}$$

$\sigma$ denotes sigmoid function, DW denotes Depthwise separable convolution. This weight matrix provides explicit filtering of the physical depth features and enables adaptive adjustment of the degree to which physical depth information is integrated into the network based on the supervision results. To achieve this, we perform element-wise multiplication of the weight matrix $\sigma$ with the physical depth features $Y_{Dep}$, resulting in the extraction of key physical depth features $Y_{Dep}^{At}$ with varying degrees of attention as determined by the network.

Since the physical depth information $Y_{Dep}^{At}$ is combined with the image features $Y_{RGB}$, we add them to the input image features $Y_{RGB}$ to generate multimodal features $Z$ enriched with the key physical depth information $Y_{Dep}^{At}$. The formula is presented below:

$$Y_{Dep}^{At} = \alpha \cdot Y_{Dep}, Z = Y_{Dep}^{At} + Y_{RGB} \tag{19}$$

multimodal features $Z$, which can be used as input to any current monocular depth estimation supervise model for image depth estimation.

| Method | Cap | AbsRel↓ | Sq Rel↓ | RMSE↓ | RMSE log↓ | $\delta < 1.25$ ↑ | $\delta < 1.25^2$ ↑ | $\delta < 1.25^3$ ↑ |
|---|---|---|---|---|---|---|---|---|
| Eigen.Eigen & Fergus (2015) | 0-80m | 0.203 | 1.548 | 6.307 | 0.282 | 0.702 | 0.898 | 0.967 |
| VNL.Yin et al. (2019) | 0-80m | 0.072 | - | 3.258 | 0.117 | 0.938 | 0.990 | 0.998 |
| BTS Lee et al. (2019) | 0-80m | 0.061 | 0.261 | 2.834 | 0.099 | 0.954 | 0.992 | 0.998 |
| PWA Lee et al. (2021) | 0-80m | 0.060 | 0.221 | 2.604 | 0.093 | 0.958 | 0.994 | 0.999 |
| Adabins Bhat et al. (2021) | 0-80m | 0.058 | 0.190 | 2.360 | 0.088 | 0.964 | 0.995 | 0.999 |
| P3Depth Patil et al. (2022) | 0-80m | 0.071 | 0.270 | 2.842 | 0.103 | 0.953 | 0.993 | 0.998 |
| DepthFormer Li et al. (2023b) | 0-80m | 0.052 | 0.158 | 2.143 | 0.079 | 0.975 | 0.997 | 0.999 |
| NeWCRFsPatil et al. (2022) | 0-80m | 0.052 | 0.155 | 2.129 | 0.079 | 0.974 | 0.997 | 0.999 |
| iDiscPiccinelli et al. (2023) | 0-80m | 0.050 | 0.145 | 2.067 | 0.077 | 0.977 | 0.997 | 0.999 |
| URCDCShao et al. (2023) | 0-80m | 0.050 | 0.142 | 2.032 | 0.076 | 0.977 | 0.997 | 0.999 |
| MiM(base) Xie et al. (2023) | 0-80m | 0.052 | 0.141 | 2.050 | 0.078 | 0.976 | 0.998 | 0.999 |
| MiM(large) | 0-80m | 0.050 | 0.139 | 1.966 | 0.075 | 0.977 | 0.998 | 0.999 |
| our(base) | 0-80m | 0.0271 | 0.0483 | 1.2301 | 0.0442 | 0.9959 | 0.9993 | 0.9998 |
| our(large) | 0-80m | 0.0251 | 0.0428 | 1.1652 | 0.0415 | 0.9966 | 0.9994 | 0.9998 |

Table 1: For a quantitative depth comparison using the Eigen split of the KITTI dataset, we employ MIM (MonoDepth3) as our supervised model. Specifically, we utilize MIM with the following configurations: MIM Base: Swin_v2_base, MIM Large: Swin_v2_large.

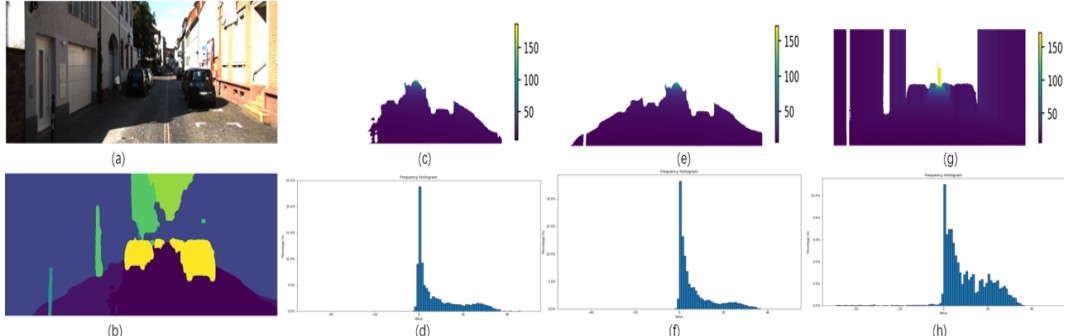

Figure 4: Extension of the Physics Depth Algorithm on sample KITTI image. (a) RGB image of the sample. (b) Semantic segmentation image of the sample. (c) Physics depth map for road pixels, along with its scale. (d) Frequency distribution of percentage errors for road pixels. (e) Physics depth map for all ground surface pixels, along with its scale. (f) Frequency distribution of percentage errors for all ground surface pixels. (g) Physics depth map for all ground surface pixels and vertical surfaces, along with its scale. (h) Frequency distribution of percentage errors for all ground surface pixels, comparing the values between physics depth and LiDAR depth.

## 4.2 SUPERVISION MODEL MODULE

In this module, we can employ any existing supervised learning model and seamlessly integrate it by directly utilizing the final features obtained from the Information Fusion Module as inputs. No structural modifications are required. Consequently, our module can be seamlessly integrated with any supervised model. Within this section, we introduce a smoothing loss function designed to enhance the accuracy of monocular depth prediction without introducing additional errors.

In this study, RGB images serve as a global prior knowledge source for applying smoothing constraints to the depth map, ensuring a smoother property for the depth or surface normal vectors. Smoothing constraints find widespread utility in depth estimation tasks, given that depth maps or surface normal vectors often exhibit noise and discontinuities. These imperfections can be attributed to various factors such as illumination variations, material disparities, occlusions, motion blur, and more within the image. By implementing a smoothing constraint, we effectively mitigate these noise and discontinuities in the depth map or surface normal vectors while endeavoring to preserve intricate details.

$$Loss(smooth) = \frac{1}{N} \sum_{i,j} |\partial_x d_{i,j}| \, e^{-\|\partial_x I_{i,j}\|} + |\partial_y d_{i,j}| \, e^{-\|\partial_y I_{i,j}\|} \qquad (20)$$

We promote local smoothness of the depth map by incorporating an L1 penalty on the depth gradients $\partial d$. Since depth discontinuities frequently coincide with image gradients, as observed in Godard et al. (2017), we augment this cost term by introducing an edge-aware factor based on the RGB image gradients $\partial I$.

## 5 EXPERIMENTS

### 5.1 SEGMENTATION FOR PHYSICS DEPTH COMPUTATION

Figure 2 illustrates the different stages of the extension of the physics depth logic,: RGB image of the sample; semantic segmentation image of the sample; The physics depth map for road pixels,

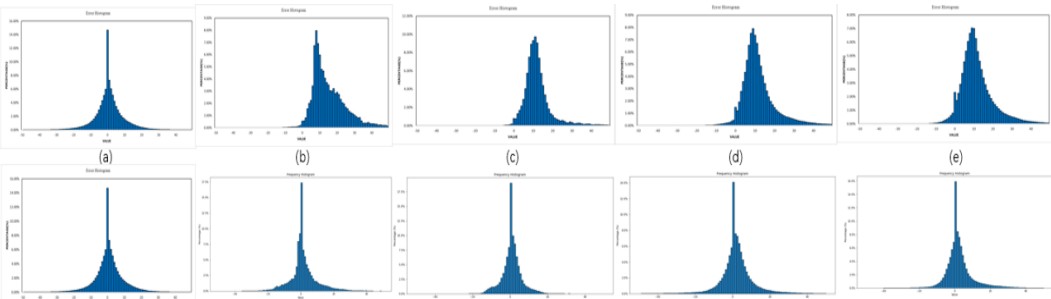

Figure 5: Percentage error frequency distribution for the entire KITTI dataset is illustrated. The top row of the figure showcases the accuracy of the physics depth logic, presenting a frequency distribution of percentage errors for road pixels across the complete KITTI dataset. This distribution enables a direct comparison between the physics depth and KITTI's LiDAR depth for the specified timestamps: (a) 2011-09-26, (b) 2011-09-28, (c) 2011-09-29, (d) 2011-09-30, and (e) 2011-10-03. The second row illustrates the analogous distributions after corrections have been applied to KITTI's camera calibration.

accompanied by its scale; The percentage error frequency distribution for road pixels; The physics depth map for the whole ground surface pixels, accompanied by its scale; The percentage error frequency distribution for whole ground surface pixels; The physics depth map for the whole ground surface pixels and vertical surfaces, accompanied by its scale, The percentage error frequency distribution for whole ground surface pixels, comparing the values between physics depth and LiDAR depth. We can see that based on semantics information, we can generate accurate physics depth values very close to the LiDAR ground truth with small errors.

## 5.2 PHYSICS DEPTH CALCULATION

Figure 4 demonstrates the effect of our physics depth calculation output for both the ground surface and regions connecting with the ground, such as vehicles and buildings. One can notice that with the camera model, the depth can be calculated very accurately for both ground regions and regions connecting the ground, both visually (top row) and quantitatively (error as shown in bottom row).

## 5.3 KITTY DATASET CORRECTION

**Testing on the Entire KITTI Dataset:** The KITTI dataset consists of five distinct calibration files, each corresponding to data collected on different days. In Figure 5, we conducted a percentage error frequency distribution analysis for each day, and the results are as follows: The error frequency histograms clearly demonstrate a substantial improvement in the performance of the physics depth algorithm after the KITTI camera calibration was corrected using LiDAR 3D depth points. This highlights the calibration inconsistencies in KITTI dataset, particularly after the initial day of data.

## 5.4 KITTY MONODEPTH EVALUATION

**KITTI Results:** Using the standard KITTI Eigen split, which comprises 697 images, we conducted an evaluation of our model. Table 1 presents a summary of the performance of state-of-the-art (SoTA) supervised methods on the KITTI dataset, clearly indicating that our method outperforms previous approaches significantly. Even in comparison to MIM (MonoDepth3) using the same model architecture, the introduction of physics depth information led to substantial improvements. Specifically, for the swin_v2_base structure, the RMSE (Root Mean Square Error) improved from 2.05 to 1.2301, and for the swin_v2_large structure, it improved from 1.96 to 1.1652. In Figure 6, we observe that when comparing the prediction results of AdaBins, SwinV2-L, 1K-MIM, and NeWCRFs models, our model excels in capturing intricate scene details and demonstrates superior scene recovery capabilities.

## 5.5 ABLATION STUDY

To thoroughly assess the impact of the proposed components in our methods on performance, we conducted detailed ablation studies on the KITTI dataset, as presented in Table 2.

**Physics Depth:** We observe from the comparison between row 1 and row 2 that the impact is substantially enhanced when the physics depth information is directly fused with the RGB information. This underscores the significant potential of physics depth in improving the predictive capabilities of supervised learning models for monocular depth estimation.

**Information Fusion:** In the comparison between Row 3 and Row 4, we observe that the information fusion module, which combines features from depth and RGB images, substantially enhances the

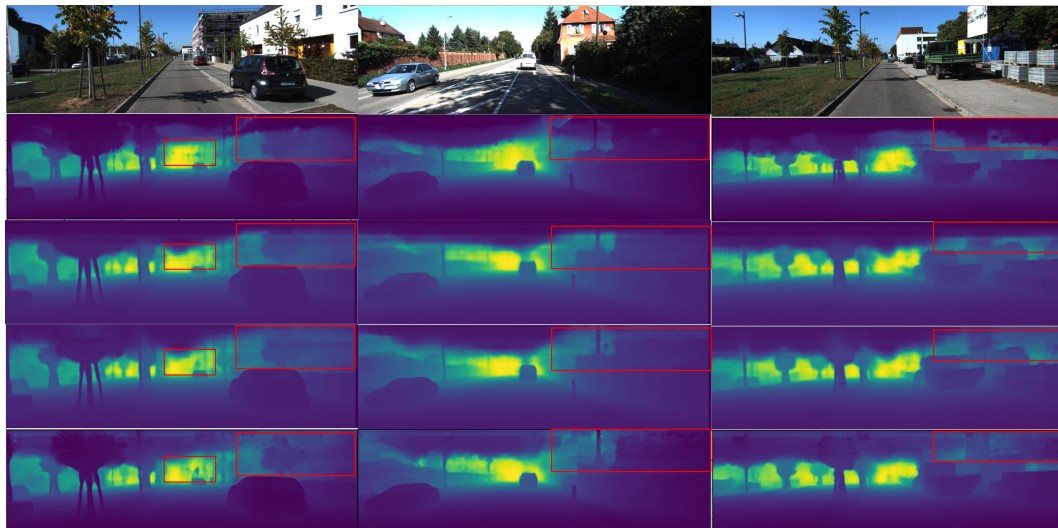

Figure 6: **Qualitative results on KITTI:**.From top to bottom the models are AdaBins, SwinV2-L 1K-MIM, NeWCRFs, our models.

| ID | PD | 80%PD | FM | AbsRel↓ | Sq Rel↓ | RMSE↓ | RMSE log↓ | $\delta < 1.25$ ↑ |
|----|----|-------|----|---------|---------|-------|-----------|-------------------|
| 1 | ✗ | ✗ | ✗ | 0.050 | 0.139 | 1,966 | 0.078 | 0.976 |
| 2 | ✗ | ✓ | ✗ | 0.0310 | 0.0620 | 1.3839 | 0.0499 | 0.9943 |
| 3 | ✓ | ✗ | ✗ | 0.0314 | 0.0648 | 1.4330 | 0.0509 | 0.9941 |
| 4 | ✗ | ✓ | ✓ | 0.0258 | 0.0453 | 1.1978 | 0.0424 | 0.9964 |
| 5 | ✓ | ✗ | ✓ | 0.0251 | 0.0428 | 1.1652 | 0.0415 | 0.9966 |

Table 2: A study of our methods on the KITTI dataset: PD: Physics Depth. IF: Information Fusion Module. 80% PD: Utilizing 80% of physics depth data

model's predictive capacity for depth estimation. Moving on to Row 4 vs. Row 5, we note that, even after employing the same information fusion module, utilizing all the data produces superior results compared to using only the top 80% of the data. Data with higher errors can still provide valuable insights to the model, whereas excessively clean data may lead to model overfitting. To address this challenge, we introduce a physics depth selection module within the information fusion module. This module intelligently highlights physics depth features that enhance model predictions while effectively filtering out features that could hinder the model's performance. This adaptive approach enables us to harness the full potential of physics depth information, leading to a significant enhancement in the model's predictive capabilities.

**80% Physics Depth:** We calculate the depth of each pixel along with its ground truth error and select the top 80% of the depth data, ordered from the smallest the largest error, as the input to our model. The results are presented in Row 2 vs. 3. It is evident that both using all the data and using the top 80% of the data enhance the predictive capabilities of the model. However, leveraging the physics depth information within the top 80% of the error range yields greater improvements. This outcome can be attributed to the fact that not all physics depth measurements are absolutely accurate, and significant errors can adversely impact the model's performance.

## 6 CONCLUSION

In this work, we introduce a physics-based supervised learning approach for depth estimation. While existing supervised techniques often rely on advanced network architectures, unique geometric priors, and diverse data augmentations to marginally enhance model performance, our method leverages physics scene depth for precise depth prediction. This significantly elevates the evaluation metrics of monocular depth estimation. By tapping into the potential of physics depth estimation calculated through camera model, we seek to enhance the performance of supervised models, especially in discerning ground and environmental details. Our ultimate aim is to enable these models to intrinsically predict accurate depth values leveraging insights from the physics depth. This approach is straightforward to deploy and offers a foundational mechanism to augment the depth prediction capabilities of various supervised models.

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
