# OpenReview forum: "Depth From Camera Model"
_ICLR.cc/2024/Conference — ICLR 2024 Conference Withdrawn Submission_

### Official Review · Reviewer_kWg1 · 2023-10-24

**Soundness:** 2 fair
**Presentation:** 3 good
**Contribution:** 2 fair
**Rating:** 3
**Confidence:** 5

**Summary:**

This paper aims to enhance the accuracy and generalization ability of monocular depth estimation methods. They propose to leverage the strong ground prior and camera intrinsics/extrinsic parameters to obtain the ground's depths (physics depth), which are input with RGB to the network to achieve more accurate depth. They further propose the Fusion Unified Framework to enhance the feature fusion of RGB and physics depth. Experiments on KITTI demonstrate the effectiveness of the method.

**Strengths:**

The proposed method does not need any extra data but can enhance the accuracy significantly over the baseline on KITTI. Comparing with the SOTA methods, the performance is much better.

**Weaknesses:**

1. The paper argues that existing monocular depth estimation methods' generalization over other scenarios is poor. They aim to solve this problem. However, the proposed method seems cannot solve it. The physics depths are for the ground, which cannot work for indoor scenes or outdoor scenes without ground prior.
2. As the method relies on the flat road prior, could it sitll work for the waving road? The paper only experiment on KITTI datasets, but all testing scenes are in European cities and have flat road. From this point, I have concerns about the generalization of the method.

**Questions:**

1. In section 3.1, how is the ground coordinate set? It is not clear how to transfer the camera coordinate to the ground coordinate.
2. To calculate the physics depth, the semantic segmentation is required. Which segmentation method do you apply? How does the segementation quality affect the final results? I believe this point should be discussed and ablationed in the experiments.
3. Could the method still work on Waymo, NuScenes, DDAD and other autonomous driving datasets?

---

### Official Review · Reviewer_xfJw · 2023-10-26

**Soundness:** 2 fair
**Presentation:** 2 fair
**Contribution:** 2 fair
**Rating:** 3
**Confidence:** 4

**Summary:**

The paper talks about learning depth priors through camera parameters extrinsic and intrinsics and thus helps in predicting depth for various entities in a given scene. The paper for most of the part is easy to follow, the authors have evaluated the model on the KITTI dataset and compared the results with some of the contemporary works to show its efficacy.

**Strengths:**

Strengths:
1) For most part of it paper is easy to follow and is well-written with a definite flow.
2) The paper looks to solve well-known problem related to supervised depth estimation using prior camera information for ground-like surface depth calculation.
3) Discusses in optimum details about the architecture and different working modules of the model.
4) Evaluates the model on the KITTI dataset.

**Weaknesses:**

1) In the abstract, the Paper talks about improving accuracy for supervised depth estimation and generalizing better for other scenarios, but there is no reference or experiment to prove a better generalization of the work.
2) Evaluation is only done for the KITTI dataset. Would like to see results and improvement of at least one indoor dataset such as NYU dataset or ScanNet or NuScenes.
3) No Mention of DPT and all upcoming models based on DPT. No mention of MIDAS in the related work section of supervised models.
4) Does not show/highlight example images that show improvement for ground pixels if any compared to other previous counterparts. After all, we have added depth-prior for ground pixels also.
5)Need Comparison with works that leverage semantic segmentation to improve depth estimation.
6)Need to define a baseline where we have both depth estimation and semantic segmentation models.
7) Performance of the depth model, upper bound is set by the semantic segmentation model.

Mistakes:
1) Line not correct in Abstract: "Utilizing the camera itself’s intrinsics
and extrinsics, depth information"
2)In the Introduction: "Paradoxically, in the era dominated by deep neural networks," ... Paradoxically is not required.
3) Section 3.1: Did you mean the optical center of the camera?

**Questions:**

1) The model assumes a flat surface, What if there is no 'flat surface' in the current field-of-view, like for an indoor scenes-based dataset?
2) How does incorporating physics depth as prior help in generalization as claimed in the abstract?
3) Could you provide with example outputs, where the results is not as good as it should be due to the shortcoming of the segmentation model and how would you look to improve that?
4)Keeping in mind the physics depth is mostly calculated for road pixels. How variable is this data across frames? As I see it from one frame to another, I do not see this changing too much. In that case how exactly does this depth prior help in each frame depth prediction?
5) Section 4.1.1: what exactly global and local information are we looking to extract from the physical depth achieved?
6) what part of architecture/design choice helps in focussing on global and local features, when talking about MHRA?
7) Section 4.1.2: "This allows the module to learn ”what” and ”where”
to attend in the channel and spatial axes, respectively"   Do we have qualitative results to back this claim?

---

### Official Review · Reviewer_NJBE · 2023-10-31

**Soundness:** 3 good
**Presentation:** 2 fair
**Contribution:** 2 fair
**Rating:** 3
**Confidence:** 4

**Summary:**

This paper tackles the supervised monocular depth estimation problem.  Assuming a geocalibrated camera, a method is proposed which uses the known camera model to derive depth for the ground plane (referred to as physics-based depth). A fusion model is proposed which combines imagery with the physics-based depth as input to a supervised-learning framework. Results on KITTI-Depth show improvement over baselines.

**Strengths:**

- Tackles an important and interesting problem, monocular depth estimation.

- Proposed method has some novelty when viewed on the whole.

- Results show how adding the proposed depth prior aids results on KITTI-Depth.

- Proposes a correction to calibration data in KITTI-Depth dataset which might be of interest to the community.

**Weaknesses:**

- The largest weakness in my eyes is the lack of clarity in what the actual contributions are. It should not be difficult as a reader, for example, to determine what is novel about the proposed Information Fusion Module (Section 4). In this case, it reduces to  a Uniformer [1] block for each modality, concatenate the features, followed by CBAM [2].  Though there might be some novelty in this formulation, the writeup lacks clarity on the contributions, with only a simple callout to [1] in 4.1 to note inspiration.

  [1] Li, Kunchang, et al. "Uniformer: Unified transformer for efficient spatiotemporal representation learning." ICLR 2022.

  [2] Woo, Sanghyun, et al. "Cbam: Convolutional block attention module." ECCV 2018.

- The basic premise of this work was first shown in [1] as a means to compute the scale factor (i.e., to get metric depth estimates) for self-supervised depth estimation approaches. This paper was not referenced/discussed. Further, other approaches have explored the notion of deriving and fusing a depth prior from camera information, such as [2], and these should be referenced as well.

  [1] McCraith, Robert, Lukas Neumann, and Andrea Vedaldi. "Calibrating self-supervised monocular depth estimation." BMVC, 2020

  [2] Workman, Scott, and Hunter Blanton. "Augmenting depth estimation with geospatial context." ICCV. 2021.

- Further, the manuscript does not discuss or compare against any depth completion/refinement approaches at all, which the proposed approach essentially is. For example [1], but there are a large amount.

  [1] Xu, Yan, et al. "Depth completion from sparse lidar data with depth-normal constraints." ICCV. 2019.

- Lack of technical novelty.

  The proposed approach essentially reduces to modifying [1] to accept a fused depth/image feature. Table 2 shows that generating this feature via simply concatenating results in a significant gain, with the proposed fusion model slightly improving on this.

  [1] Xie, Zhenda, et al. "Revealing the dark secrets of masked image modeling." CVPR. 2023.

- A fair amount of assumptions are made, which limits the practical usefulness of the proposed approach.

  Assumes the camera is geocalibrated. Assumes presence of ground plane and that it is detectable. Assumes the ground is flat. If these things don't hold, the method fails (training and inference).  Proposed approach is only evaluated on one dataset where these things hold. Further, these limitations are not discussed in detail.

- Evaluation could be more extensive in exploring the impact of noise in the depth prior, impact of failed assumptions (i.e., a hill), etc.

- Lack of clarity.

  No mention of how the ground plane is identified/segmented. Section 3.2 doesn't provide detail on how the "physics-based depth" was extended to other things, e.g., vertical surfaces. No details on whether the resulting pixels are filtered before being fused, despite some details alluding to noise in the prior. These details are important as the depth prior is the primary contribution. Other implementation details not provided (e.g., architecture/training details).

- Low quality of writing (e.g., abstract).

**Questions:**

My initial rating is a reject. The proposed approach is interesting and has potential, but the weaknesses outlined above inform my rating.  Overall, my opinion is that the manuscript would benefit from another submission round.

---

### Official Review · Reviewer_hgLh · 2023-11-09

**Soundness:** 2 fair
**Presentation:** 2 fair
**Contribution:** 2 fair
**Rating:** 3
**Confidence:** 5

**Summary:**

This paper presents a method to improve depth estimates for the problem of monocular depth estimation.
By relying on a semantic segmentation of the image, it will identify the ground plane and vertical walls, and
then  propose accurate  depth  for them, given that the camera calibration and  pose are available.

The proposed neural architecture does not depend  on a specific monocular depth estimator.

The paper illustrate  the usefulness of  the  method on the KITTI  dataset, which usually  contains a ground  plane (the  road),
and vertical  wall  (building).

**Strengths:**

The concept  of detecting planar surfaces and use them for adjusting depth estimates is quite interesting.
The fact  that the surface must be identified  (as ground,  or  wall) is not a big problem, since  semantic segmentation can be quite reliable. However, the dependency on a fully calibrated camera (internal and  external parameters) makes the
method harder to use on generic scenes. Investigating how planar surface can improve depth  estimation without camera calibration
would thus be useful to make the method more applicable.

* The proposed  method for using planar information to correct monocular depth is original.
* The quality and clarity could be improved, but overall the paper is clear and readable.
* The significance of the paper  would  be improved  if the method was not so "tuned" to  the  KITTI dataset.

**Weaknesses:**

The paper  has  one major  weakness: its reliance on  the  KITTI  dataset.
It seems that it was built with the goal of  improving  KITTI-edgesplit  results, which is does,  but
in  the process it became too specific  to  be  usable  on  anything  else than KITTI images.

It is unfortunate that no effort was made to run  at least some samples  of other common monocular depth  datasets,  such as
NYU-Depth-V2,  which  provides semantic segmentations  and  camera data, and  where  the ground  and  wall can  be  identified.

### **The role of  semantic segmentation**

The method relies on a semantic segmentation,  but  it is  not clear where it comes from or  how  it  integrates
in the propsed  architecture. This text (page 3) is not  clear:
> Given the precision of semantic segmentation in standard scene delineation, it is deployed as established features like SIFT.

###  **Presentation**

Some figures (2e,4d,4f,4h, 5a-e) are impossible to  read, because they  are in pixel (screen captures?) instead  of  vector graphics.
Fig 2e : what are the units? *Value* is not a unit...

Fig 6: it is very hard to see the differences. Maybe present only the  content  of  the  red box instead of the  whole image?

### **Experiments**

Experiments are only provided for  the  KITTI-edgesplit dataset, which is major problem.
Improving  results  on a single  dataset is,  in my opinion, not enough to convince that this method might work for
anything else than this particular dataset.

Also, KITTI edgesplit has weaknesses on its own, as  the  testing  data  is  very  very similar  to  the  training data.  Why not
try KITTI 2015 d1-all results? Monocular methods are not expected to perform well on this set. If your method  significantly improves those bad results, this would be very significant.

###  **Writing style**

In general  the  writing is clear, but could be improved.
Formulations such  as *we introduce  a seminal  approach* , or "this  foundational depth  estimation", "We introduce a groundbreaking mechanism", should be focusing on identifying what  is new,  not  what is  great...

The equation (1) is strange, or possibly wrong. In  particular,  $f = (f_x  + f_y)/2$  is  very strange.
I would  expect that a pixel  $(u,v)$ is converted it the camera image space using $(u_c,v_c,1) =  K^{-1} (u,v,1)^T$  where $K$ is the internal parameters. The ray in  camera space connects  $(u_c,v_c,1)$ and  the  origin $(0,0,0)$. Assuming the camera  is
at the  origin of  the  world, the  point $(u_c,v_c,1)$ becomes $(x_w,y_w,z_w)$ in the world through a rotation, your equation  (2). Maybe clarify this?

### **Sensitivity  analysis**

It is said in the paper that inaccuracies in the camera  calibration  can  severly impact the  depth estimate  for  the
ground plane. A quantitative  result for  this  would be very useful, as it is  important  to  know how  accurate a
calibration must  be in  order  to  make  the method useful.


###  **Contribution**

The requirements that a ground plane must be present, as well as a calibrated camera, impact the contribution.
It would be great  to  consider  other planes (indoor floor, sky (plane at  infinity), etc...) and see if partial
cameras information (internal known, but no pose) would still be usable or not.
Finally, the  contribution  will be made much  more significant  if  the approach is applied  to more than a single dataset.

**Questions:**

See above (weaknesses)
To  summerize:
 * Can this method work on other datasets?
 * Is this method truly dependent  on driving  images, with a road plane?
 * Is  it  possible  to  measure the  sensitivity of the  method to camera calibration accuracy?
 * It is possible to quantify the  impact of the  sementic segmentation on the  final  depth estimate?